# Observing mesoscale dynamics with multistatic specular meteor radars: first climatology of momentum flux, horizontal divergence and relative vorticity over <u>northern</u> central Europe

J. Federico Conte<sup>1</sup>, Jorge L. Chau<sup>1</sup>, Toralf Renkwitz<sup>1</sup>, Ralph Latteck<sup>1</sup>, Masaki Tsutsumi<sup>2</sup>, Christoph Jacobi<sup>3</sup>, Njål Gulbrandsen<sup>4</sup>, and Satonori Nozawa<sup>5</sup>

**Correspondence:** J. Federico Conte (conte@iap-kborn.de)

**Abstract.** Continuous and reliable measurements of the mesosphere and lower thermosphere (MLT) are key to further the understanding of global atmospheric dynamics. Observations at horizontal scales of a few hundred kilometers (i.e., mesoscales) are particularly important since gravity waves have been recognized as the main drivers of various global phenomena, e.g., the pole-to-pole residual meridional circulation. Multistatic specular meteor radars are well suited to routinely probe the MLT at these scales. One way to accomplish this, is by investigating the momentum flux, horizontal divergence  $(\nabla_H \cdot \mathbf{u})$  and relative vorticity  $((\nabla \times \mathbf{u})_z)$  estimated from the Doppler shifts measured by a radar network. Furthermore, the comparison between the horizontal divergence and the relative vorticity can be used to determine the relative importance of gravity waves (i.e., divergent motions) and strongly stratified turbulence (i.e., vortical motions). This work presents the first climatology of all these estimates together, as well as results on the probability distribution of the total momentum flux (TMF), and the comparison between  $\nabla_H \cdot \mathbf{u}$  and  $(\nabla \times \mathbf{u})_z$ , obtained from almost 10 years of continuous measurements provided by two multistatic specular meteor radar networks: MMARIA/SIMONe Germany, covering an area of more than 200 km radius around (53°N, 11°E), and MMARIA/SIMONe Norway, which covers an area of similar size, but around (69°N, 16°E). Among others, our results indicate that at middle latitudes the horizontal divergence and the relative vorticity are balanced around summer mesopause altitudes, while the former dominates over the latter above  $\sim$ 90 km of altitude during parts of the fall transition. At high latitudes, the vortical motions dominate during late spring and early summer. Besides, the strongest 5% of GWs contribute much more over northern Germany than over northern Norway, where the larger values of the excess-kurtosis indicate that the contribution from the small-amplitude GWs is also more significant at middle latitudes, especially during the summer. In other words, the TMF in the mesosphere and lower thermosphere over central Europe is considerably more intermittent at middle latitudes than at high latitudes.

<sup>&</sup>lt;sup>1</sup>Leibniz-Institute of Atmospheric Physics at the University of Rostock, Kühlungsborn, Germany

<sup>&</sup>lt;sup>2</sup>National Institute of Polar Research, Tokyo, Japan

<sup>&</sup>lt;sup>3</sup>Institute of Meteorology, Leipzig University, Leipzig, Germany

<sup>&</sup>lt;sup>4</sup>University of Tromsø- The Arctic University of Norway, Norway

<sup>&</sup>lt;sup>5</sup>Nagoya University, Nagoya, Japan

#### 20 1 Introduction

The vertical flux of horizontal momentum, the horizontal divergence and the relative vorticity are fundamental parameters to describe the dynamics of the terrestrial middle atmosphere, particularly at horizontal scales of a few hundred kilometers (i.e., mesoscales). Fundamental because divergent and vortical motions with these scales can significantly impact the transport of energy and momentum, and because depending on which of the two types of motion dominates, it is possible to respectively determine whether the atmospheric motions are driven by gravity waves (GWs) or layered anisotropic stratified turbulence (LAST), although a balance between these two is most likely the general case (Lindborg, 2007; Brethouwer et al., 2007; Avsarkisov, 2020; Avsarkisov and Conte, 2023).

Gravity waves are atmospheric buoyancy oscillations with periods spanning from some minutes to several hours, with corresponding horizontal scales of a few to several hundred kilometers, and vertical wavelengths typically less than ~15 km (e.g., Fritts and Alexander, 2003; Baumgarten et al., 2017). GWs are ubiquitous and can be generated by many sources or processes, including the orography, deep convection, wave-breaking, shear instabilities and non-linear interactions (e.g., Gardner, 1994; Sato et al., 2009). The crucial role that GWs play in driving and contributing to global atmospheric phenomena, such as the residual pole-to-pole meridional circulation and the quasi-biennial oscillation, has already been recognized for several decades (e.g., Fritts, 1984; Garcia and Boville, 1994; Becker, 2012). However, only recent studies have suggested that the rotational component of the mesoscale wind fields (associated with vortical motions) may contribute to the energy budget as significantly as the divergent motions, which are associated with internal GWs (e.g., Caulfield, 2020, 2021). Vortical motions are driven by turbulent processes. However, given the strong vertical stratification of the middle atmosphere, a new theory that could reconcile turbulence with anisotropy was needed. This theory, called layered anisotropic stratified turbulence (Billant and Chomaz, 2001; Lindborg, 2006), has recently been used to show that strongly stratified turbulence might be as significant as gravity waves in driving mesoscale processes in the mesosphere and lower thermosphere (MLT) region (Avsarkisov and Conte, 2023).

Gravity waves drive atmospheric motions by transporting and redistributing momentum in the mean background flow. Thus Hence, the importance of estimating the GW horizontal momentum and determining how it is transported with height, i.e., its vertical flux. On the other hand, the ability of the GWs to reach the right altitudes necessary to altitudes at which they can influence atmospheric processes depends on the background flow conditions. This means that any changes in the background winds have a direct impact on the upward propagation of the gravity waves. These wind variations can result, e.g., from the breaking of other atmospheric waves. Among them, tides play an important role due to the large amounts of energy they transport, as well as the considerable amplitudes they can reach, particularly at MLT altitudes (e.g., Oberheide et al., 2009; Akmaev et al., 2016). Thermal tides are planetary-scale oscillations generated mainly by the absorption of solar UV and infrared radiation by ozone in the stratosphere and water vapor in the troposphere, respectively (e.g., Forbes, 1984; Hoffmann et al., 2007; Pancheva and Mukhtarov, 2011). They have periods that are sub-harmonics of a solar day, with the semidiurnal (or 12-hour) and diurnal (or 24-hour) tides being the dominant modes (e.g., Murphy et al., 2006; Conte et al., 2018). Tides can

also be generated by latent heat release, gravitational forces and non-linear interactions (e.g., Hagan and Forbes, 2002; Forbes and Zhang, 2012; He et al., 2017).

55

65

Layered anisotropic stratified turbulence is the result of complex interactions between different types and scales of motion, under the influence of strong vertical stratification. It has already been recognized in the context of synoptic-scale atmospheric and mesoscale oceanic flows (e.g., Lindborg, 2006; Riley and Lindborg, 2013; Falder et al., 2016). However, its role in atmospheric mesoscale dynamics and its connection with GWs has gained attention only recently (e.g., Avsarkisov et al., 2022; Poblet et al., 2023). Little is known in this matter, although a few recent studies suggest a redistribution of energy across different scales by means of pancake-like thin turbulent structures that spread with the horizontal background wind (e.g., Caulfield, 2020, 2021; Avsarkisov and Conte, 2023). Among these few studies, perhaps only a couple have used observations, and just on a campaign basis. Thus, continuous measurements of dynamical parameters that can capture the mesoscale variability, at least on average, are essential to determine the sources driving these changes, particularly at MLT altitudes, where the energetic transition from synoptic- to mesoscales is not fully understood.

Multistatic specular meteor radars (MSMRs) can contribute considerably to this important goal. These are systems that operate continuously, utilizing multiple transmitting and receiving stations to observe meteor trails across a wider geographic area, allowing for a more comprehensive assessment of the winds and their associated dynamical properties (e.g., Stober and Chau, 2015; Chau et al., 2019). Conventional specular meteor radars, i.e., monostatic SMRs, have been used considerably to investigate momentum fluxes in the MLT region (e.g., Hocking, 2005; Fritts et al., 2010a, b; Placke et al., 2011a, b; Andrioli et al., 2015; de Wit et al., 2016, 2017). However, estimates of the momentum flux obtained from SMR measurements must be carefully interpreted, as they are adversely affected by the correlated errors, unresolved geophysical variability and some limitations inherent to the observational geometry (e.g., Kudeki and Franke, 1998; Conte et al., 2021). By detecting much larger amounts of meteors, and more importantly, meteors that are distributed more homogeneously in space and time, MSMRs can reduce the adverse impact of these issues, and thus provide more reliable momentum flux estimates (e.g., Spargo et al., 2019; Conte et al., 2021; Charuvil Asokan et al., 2022). Furthermore, the measurements made by MSMRs can be used to estimate the relative vorticity and the horizontal divergence of the horizontal winds (e.g., Chau et al., 2017; Stober et al., 2018). A simple comparison between these two parameters can be of great help to routinely assess the relative importance of vortical and divergent motions at altitudes between 80 and 100 km.

In recent years, the Leibniz-Institute of Atmospheric Physics (IAP), in Kühlungsborn, Germany, has deployed several MSMR networks in different parts of the world (e.g., Conte et al., 2022; Poblet et al., 2023; Conte et al., 2023). Particularly in northern central Europe, the IAP has been operating the two oldest MSMR networks: one in northern Norway for more than a decade, and another one in northern Germany for over seven years. Thus, having relatively long sets of almost uninterrupted measurements made by these two meteor radar networks constitutes an excellent opportunity to present the first climatology of momentum fluxes, horizontal divergence and relative vorticity observed in the MLT over middle- and high-latitude sectors. The rest of this paper is structured as follows. The next section briefly describes the instrumentation and the data analysis techniques applied to the measurements provided by the meteor radar networks. The results are presented in Section 3 and discussed in Section 4. Finally, in Section 5 the concluding remarks of this work are stated.

## 2 Instruments and data processing

100

105

110

As mentioned in the previous section, the purpose of this study is to provide the first climatology of neutral mesoscale dynamics at MLT altitudes (~80-100 km) over middle and high latitudes in Europe. For this reason, two MSMR networks have been considered: MMARIA/SIMONe Germany, which integrates SIMONe Germany with the pulsed SMR in Collm (near Leipzig) and two remote receiving stations, and MMARIA/SIMONe Norway, which comprises SIMONe Norway and the pulsed SMRs in Tromsø and Alta, with a recent expansion to include the Kiruna SMR, in Sweden. MMARIA stands for Multistatic Multifrequency Agile Radar for Investigations of the Atmosphere, and it summarizes the concept of integrating multiple meteor radar systems to investigate the MLT over a given place on Earth, e.g., northern Germany. This concept was developed at the IAP about one decade ago (Stober and Chau, 2015), and can be implemented with both pulsed and coded continuous wave meteor radar systems (e.g., Chau et al., 2019; Conte et al., 2022). The latter approach has been named SIMONe (Spread spectrum Interferometric multistatic Meteor radar Observing Network), and it has been proven to provide more reliable wind estimates over larger volumes, and to be significantly more cost-effective and easier to install and expand than conventional pulsed SMRs (e.g., Vierinen et al., 2019; Charuvil Asokan et al., 2022). Independently of the transmitting technology, MSMRs have the advantage of detecting more meteors and increasing the viewing-angle diversity of the observed volume. The latter advantage is crucial to implement approaches such as VVP (Volume Velocity Processing), which allow for the estimation of key parameters like the relative vorticity (e.g., Waldteufel and Corbin, 1979; Chau et al., 2017). Currently, both networks can be monitored online at http://maarsv.rocketrange.no/MMARIA/. For more details, please see Poblet et al. (2023) or contact the persons mentioned in the given link.

To investigate the mean annual dynamics (or climatology) of the MLT over northern Germany and northern Norway, several atmospheric parameters were estimated from the measurements provided by the MSMR networks mentioned above. Namely, the mean horizontal winds, the horizontal wind gradients, and the vertical fluxes of zonal ( $\rho < u'w' >$ ) and meridional ( $\rho < v'w' >$ ) momentum. To estimate all these parameters, three variants of the same equation were solved by means of least squares fitting. First, the mean horizontal winds were estimated considering (e.g., Hocking et al., 2001; Holdsworth et al., 2004)

$$2\pi f + \eta = \mathbf{u} \cdot \mathbf{k} = (u, v, 0) \cdot (k_x, k_y, k_z) = uk_x + vk_y, \tag{1}$$

where u and v are the mean zonal (east-west) and meridional (north-south) wind components, respectively (the mean vertical wind, w, is assumed to be equal to zero). Besides,  $\mathbf{k} = (k_x, k_y, k_z)$  is the Bragg wave vector (scattered minus incident wave vectors) in the meteor-centered east-north-up coordinate system (perpendicular to the meteor trail), f is the Doppler shift and  $\eta$  is the Doppler shift uncertainty. These two wind estimates were calculated in time-altitude running bins of 1-hour, 2-km, shifted by 1 hour and 1 km, respectively. In other words, u and v are hourly mean winds with a height resolution of 2 km.

To obtain the horizontal wind gradients, the same Equation 1 was considered, but with the difference that the wind vector  $\mathbf{u}$  is instead given by a first order Taylor expansion in the horizontal components,

$$2\pi f + \eta = \mathbf{u} \cdot \mathbf{k} = \begin{pmatrix} u_0 + \frac{\partial u}{\partial x} \Delta x + \frac{\partial u}{\partial y} \Delta y + \frac{\partial u}{\partial z} \Delta z, \\ v_0 + \frac{\partial v}{\partial x} \Delta x + \frac{\partial v}{\partial y} \Delta y + \frac{\partial v}{\partial z} \Delta z, \\ w_0 \end{pmatrix} \cdot \begin{pmatrix} k_{x\_}, k_{y\_}, k_z \end{pmatrix}.$$
 (2)

The derivatives with respect to x, y and z correspond to the changes in the zonal, meridional and vertical directions, respectively. The displacements  $\Delta x$ ,  $\Delta y$  and  $\Delta z$  are calculated considering the World Geodetic System 1984, or WGS84, (see appendix A in Stober et al., 2018). The reference point to determine the displacements locates at  $\mathbf{p}_G = (53.98^{\circ}\text{N}, 11.7^{\circ}\text{E})$  for northern Germany, and  $\mathbf{p}_N = (69.3^{\circ}\text{N}, 16.04^{\circ}\text{E})$  for northern Norway. Given the larger amount of unknowns, more meteor detections are needed for a reliable estimation, and thus Equation 2 was solved in time-altitude running bins of 4-hour, 4-km, shifted by 1 hour and 1 km, respectively.

By linearly combining the estimates of the horizontal wind gradients, the following parameters were also calculated: the horizontal divergence,  $\nabla_H \cdot \mathbf{u} = \frac{\partial u}{\partial x} + \frac{\partial v}{\partial y}$ , the relative vorticity,  $(\nabla \times \mathbf{u})_z = \frac{\partial v}{\partial x} - \frac{\partial u}{\partial y}$ , and the stretching  $(\frac{\partial u}{\partial x} - \frac{\partial v}{\partial y})$  and shearing  $(\frac{\partial u}{\partial y} + \frac{\partial v}{\partial x})$  deformations. It is thanks to the multistatic configuration that both the relative vorticity and the shearing deformation can be estimated (e.g., Reid, 1987).

Finally, the vertical momentum fluxes were obtained after solving (Hocking, 2005)

$$(\mathbf{u}' \cdot \mathbf{k})^2 = (2\pi (f - \hat{f}))^2. \tag{3}$$

Here,  $\mathbf{u'} = (u', v', w')$  represents the residual wind vector, and  $\hat{f}$  is a so-called mean Doppler shift,  $\hat{f} = \mathbf{u} \cdot \mathbf{k}/2\pi$ , where  $\mathbf{u}$  is the mean wind that results from solving Equation 1, but in time-altitude running bins of 4-hour, 4-km. For consistency, Equation 3 was also solved in time-altitude running bins of 4-hour, 4-km, shifted by 1 hour and 1 km, respectively. Although the unknowns in Equation 3 are the six components of the stress tensor (divided by the air density), i.e.,  $\langle u'u' \rangle$ ,  $\langle v'v' \rangle$ ,  $\langle u'w' \rangle$ ,  $\langle v'w' \rangle$ ,  $\langle v'w' \rangle$ , and  $\langle w'w' \rangle$ , in this study we only present and discuss results on  $\langle u'w' \rangle$  and  $\langle v'w' \rangle$ .

In all three cases (Equations 1, 2 and 3), only the meteor detections with zenith angles less than  $60^{\circ}$  and that fall within an area of 230 km radius around the reference points  $\mathbf{p}_G$  and  $\mathbf{p}_N$  were taken into consideration. In addition, a first wind estimation was carried out to remove potential outliers. This implies solving Equation 1 in bins of the same size as that used later on in each case and removing those meteor detections with Doppler shifts whose values have a corresponding residual of more than 3 standard deviations (e.g., Spargo et al., 2019). The Bragg wave vector was always calculated considering the WGS84 (Chau and Clahsen, 2019). The air density ( $\rho$ ) values were taken from the US Standard Atmosphere 1976.

The datasets investigated hereinafter span from 1 January 2018 until 31 December 2024, for MMARIA/SIMONe Germany, and from 1 January 2015 until 31 December 2024, for MMARIA/SIMONe Norway.

#### 145 3 Results







Most of the results are presented in separate figures for each location. Only the results about the total momentum flux (TMF) are presented in one figure including the data from both MSMR networks. First, results on the planetary-scale dynamics are introduced, i.e., on the mean winds and tides, to then proceed with results about the mean mesoscale dynamics, i.e., the horizontal wind horizontal gradients and the momentum fluxes. Strictly speaking about the data, the only differences between the two meteor radar systems lie in the different observational periods covered by each MSMR network and in that the few existing data gaps correspond to different time intervals.

## 3.1 Spectra of hourly winds

Figure 1 shows the spectra of the horizontal winds over northern Germany. The left and right columns are used for the zonal and meridional components, respectively. The upper panels present the spectra of the hourly winds, while the lower panels show the spectra of the same hourly winds, but after subtracting one-week mean wind and diurnal and semidiurnal tidal peaks. Similarly to previous studies, the mean winds and the two tidal waves were estimated by fitting the hourly winds with a truncated Fourier series considering a running bin of 7 days and 2 km, shifted by 1 day and 1 km, respectively (e.g., Murphy et al., 2006; Hoffmann et al., 2010; Conte et al., 2019). Before applying the Fast Fourier Transform (FFT) with a Hann window, the winds were averaged over approximately one scale height around the maximum of meteor detections, i.e., between 86 and 93 km of altitude. Once the spectra were determined, a straight line (in red) was fitted by means of least squares using all spectral points between the inertia and Nyquist periods, i.e., between 15.02 and 2 hours, respectively. To assess the variability in the estimated slope, the fitting procedure was repeated 50 times, but considering a random selection of only 20% of the points. This is shown with the yellow shaded area. Besides, the The interval encompassing the 50 slope values is indicated in the legend box of each plot. Notice that the variability is in the order of 10 % or less.

Similarly, Figure 2 shows the spectra of the hourly winds before and after removing 7-day mean winds and diurnal and semidiurnal tidal peaks, but over northern Norway. The same description as that of Figure 1 applies to this second figure, except that: a) the inertia period for northern Norway is equal to 12.85 hours; b) the observational period covers three more years, starting on 1 January 2015 (instead of 2018); and c) a few gaps were found in the wind data (13 hourly values in total, not consecutive) and filled by means of linear interpolation. Notice that, as in the case of northern Germany, the variability in the slope of the fitted straight line is also less than 10 %.

A brief inspection of Figure 1 reveals that tidal waves dominate the spectrum of both wind components over northern Germany. Particularly, the semidiurnal (S2) tide exhibits the highest values, followed by the diurnal (D1) and terdiurnal (T3, or 8-hour) tides, with comparable amplitudes between them but approximately one order of magnitude smaller than that of the S2 tide. The latter is in accordance with previous studies reporting that the S2 tide dominates at middle and high latitudes (e.g., Manson et al., 1999; Hoffmann et al., 2010; Yamazaki et al., 2023). Considerable amplitudes are also observed in the tidal waves with shorter periods, namely, the quarterdiurnal (6-hour), the 4.8-hour and the 4-hour tides. The D1 and S2 tides present larger values in the meridional component, while the remaining tidal waves are slightly stronger in the zonal component,

particularly in the case of the 4-hour tide. Despite being multistatic, MMARIA/SIMONe Germany and MMARIA/SIMONe Norway observations cover an area of only a few degrees in longitude. Consequently, from these observations alone it is not possible to resolve different horizontal wave numbers for the tides. However, previous studies have investigated the forcing mechanisms of the terdiurnal tide, arriving to the conclusion that the main forcing mechanism is the solar radiation absorption by water vapour and ozone (e.g., Lilienthal et al., 2018). Thus, it is reasonable to expect that the observed terdiurnal tide is, on average (or climatologically), driven by the migrating component, and therefore has a dominant wave number equal to 3. Similarly, previous studies based on observations and model simulations have shown that the semidiurnal tide at middle and high latitudes is dominated by the migrating component (e.g., Chau et al., 2015; He and Chau, 2019), Besides, notice that the subtraction of a 7-day estimate of the semidiurnal tide from the hourly winds resulted in a clean removal of that frequency peak, keeping all other frequencies around it (see lower panels of Figure 1). This can be possible only if the phase of that tidal wave is stable and remains constant during at least seven days, which in turn suggests the dominance of the migrating semidiurnal component. Finally, notice that in the case of the diurnal tide, the energy contribution from other close frequencies was also removed after subtracting 7-day estimates of the tides. This indicates variability in the phase of the diurnal tide, even within intervals of seven days, which is interpreted as this tidal wave being driven not only by the migrating tide but also by non-migrating components, most likely DE3 (eastward-propagating wave number 3) and DW2 (westward-propagating wave number 2), modes already reported as the dominant non-migrating diurnal tides (e.g., Oberheide et al., 2011; Suclupe et al., 2023).








The horizontal wind spectra over northern Norway are also characterized by strong tidal peaks, with the semidiurnal tide exhibiting the highest values (see Figure 2). However, the tidal waves with shorter periods, including the terdiurnal tide, exhibit weaker amplitudes than over northern Germany. Furthermore, the 4.8-hour tide is not observed and the 4-hour tide is visible only in the zonal direction. Given its shorter horizontal wavelength, it can be that the These features are consistently observed across many days and years. In the case of the 4-hour tidebehaves more like a linear wave. In that sense, a short period, the spectral peak remains above the noise level and exhibits phase coherence (not shown here), suggesting that it represents an actual tidal wave rather than an artifact arising from daily variability in data quality. Furthermore, the errors in the mean winds and tidal estimates are comparable between both components, with no indication that one component (either in the mean winds or in the tides) is noisier than the other. In addition, the multistatic configuration allows for a more homogeneous distribution of meteor counts throughout the day, and the hourly detection rates are sufficiently uniform in the horizontal plane to reliably extract the tides without favoring any specific direction. Based on these considerations, it is concluded that the short-period tidal peaks are indeed actual tides and not artifacts introduced by data quality issues. However, understanding, for instance, why a short-period tidal wave propagating in the zonal direction may have exhibit a much smaller meridional component. However, tidal structures are affected by boundary conditions, making the relationship between the zonal and meridional components more complex, is left for future investigations.

Similarly to northern Germany, the diurnal and semidiurnal tides have slightly stronger values in the meridional direction. However, contrary to what is observed at middle latitudes, where the terdiurnal and quaterdiurnal tides have more energy in the zonal component, at high latitudes these two tides present slightly larger amplitudes in the meridional direction, indicating a change in the polarization of these tides as the latitude increases. The different values of the slopes shown in the upper panels (about 10% less steep compared to northern Germany) are attributed mostly to a change in the Coriolis parameter. The spectra presented in both Figures 1 and 2 comprise the energy of all the structures with associated with all structures having horizontal scales of ~400 km or more. Thus, they definitely include structures As such, they include motions that are influenced by the Coriolis effect. Notice that Notably, removing the mean wind and the dominant tides does not modify significantly significantly modify the values of the slopes at each given individual location, but it does make them the slopes more comparable between the bottom panels of both figures(-2.82 vs. -2.89 the two figures: -2.82 vs. -2.89 for the zonal componentand -3.02 vs. -2.93 and -3.02 vs. -2.93 for the meridional one). In other words, after the removal of the two dominant perturbations that can be influenced by the Coriolis effect (the S2 and D1 tides), despite the remaining tidal peaks, most of the energy corresponds to shorter period tidal peaks and a reduction in the total energy (compared to the full hourly wind spectrum), the remaining dynamics appear to be dominated by structures with horizontal scales where for which the Coriolis effect has little to no impact (i.e., mesoscales), making the leading to the similar spectral slopes between middle and high latitudes very similar. This shows the consistency of our results with previous studies suggesting that the MLT energy spectrum for periods shorter than the inertia period is dominated by mesoscale gravity waves and a strongly stratified turbulence regime (e.g., Lindborg, 2006; Vierinen et al., 2019).

# 3.2 Vertical flux of horizontal momentum, horizontal divergence and relative vorticity

To further investigate the MLT mean dynamics at mesoscales over northern Germany and northern Norway, Figures 3 and 4 respectively present the climatology of 28-day median values of the momentum fluxes and six other parameters. Specifically, from top to bottom, (left) the vertical flux of zonal momentum ( $\rho < u'w' >$ ), the horizontal divergence ( $\nabla_H \cdot \mathbf{u}$ ) and the stretching deformation ( $u_x - v_y$ ); and (right) the vertical flux of meridional momentum ( $\rho < v'w' >$ ), the relative vorticity (( $\nabla \times \mathbf{u}$ )<sub>z</sub>) and the shearing deformation ( $u_y + v_x$ ) are shown. The mean background conditions are shown in the bottom panels with the mean (left) zonal and (right) meridional winds. The dash-dotted blue contour lines bound the areas of regions with highest year-to-year variability, i.e., the areas with the largest values of where the  $2\sigma$ , parameter (with  $\sigma$  being denoting the standard deviation) reaches the highest values. For the purposes of an easier comparison, the colour bars are the same for northern Germany (Figure 3) and northern Norway (Figure 4). The momentum flux estimates are in units of mPa and the mean winds in  $ms^{-1}$ . All other parameters are presented in units of  $ms^{-1}km^{-1}$ .

Previous studies have shown that in order to reduce the statistical uncertainties in the momentum fluxes estimated from SMRs observations, these estimates must be averaged over long intervals of time: at least 25 days at MLT altitudes (e.g., Kudeki and Franke, 1998; Conte et al., 2021). It is for this reason that the momentum flux estimates presented in Figure 3 and Figure 4 are 28-day median values of the 4-hour, 4-km values described in the previous section. Besides, to make proper comparisons between the different dynamical parameters investigated in this work, the other quantities presented in Figures 3 and 4 also correspond to 28-day median values of the 4-hour, 4-km estimates calculated following the procedure described in Section 2. The reason to use the median (instead of the mean), and to compute it over 28 days, is consistency with previous works (Conte et al., 2022, 2023).

#### MMARIA Germany (1.1.2018-31.12.2024)

**Figure 1.** Wind spectra over northern Germany. The left and right columns correspond to the zonal and meridional components, respectively. The upper panels show the spectra of hourly winds estimated between 1 January 2018 and 31 December 2024. The lower panels present the spectra of the hourly winds after subtracting 7-day mean winds and diurnal and semidiurnal tides. The relative total signal energy (rTSE) is indicated for two cases: (light blue) the entire spectrum and (light orange) only the part of the spectrum corresponding to periods shorter than the inertia period. These values are expressed as percentages of the normalized total signal energy (nTSE), that is, the total energy under the full frequency spectrum of the hourly wind time series. The red straight lines were fitted by least squares considering all spectral values between the inertia and Nyquist periods (15.02 and 2 hours, respectively). The yellow shaded area indicates the variability in the slope, whose range of values is indicated between parentheses in the legend box (see text for details).

To assess the relative importance between mesoscale waves and strongly stratified turbulence, and also have a better picture of how big volumes of air move at MLT altitudes, Figure 5 and Figure 6 respectively present for northern Germany and northern Norway, the difference in absolute value between the composite values of the horizontal divergence and the relative vorticity, the stretching deformation and the shearing deformation, and the composites of each individual horizontal wind horizontal gradient. From top to bottom, on the left,  $\nabla_H \cdot \mathbf{u}$ ,  $(u_x - v_y)$ ,  $u_x$  and  $v_y$ ; in the middle,  $(\nabla \times \mathbf{u})_z$ ,  $(u_y + v_x)$ ,  $u_y$  and  $v_x$ ; and on the right, the difference (in absolute value) of the parameters on the left column minus those in the middle one. All parameters are in units of  $ms^{-1}km^{-1}$ . To be consistent with the results presented in Figures 3 and 4, 28-day median values of each parameter were used to calculate the corresponding composite values.



Figure 3 shows that the vertical flux of zonal momentum during the northern hemisphere's winter and early spring is negative below  $\sim$ 90 km and positive above. Assuming that when averaged over 28 days the GW propagation is mostly upward directed, this means that during the winter and early spring the zonal momentum flux at middle latitudes over central Europe is westward

below 90 km of altitude and eastward above. The rest of the year is dominated by an upward flux of westward zonal momentum above  $\sim$ 86 km, while below that altitude  $\rho < u'w' >$  alternates between eastward and westward directions. At high latitudes (Figure 4) the zonal momentum flux behaves in accordance with the residual meridional circulation, that is, the summer shows a strong eastward zonal momentum flux around 81 – 82 approximately between 83 and 86 km of altitude that progressively, that then decreases its intensity as the altitude increases, to eventually become westward directed in the lower thermosphere (above 95–96 km of altitude). The rest of the year, the zonal momentum flux over northern Norway is mostly westward directed, except during winter and early spring below 81 km of altitude, where  $\rho < u'w' >$  is mainly eastward.







The horizontal divergence over northern Norway (see second panel on the left in Figure 4) also behaves in accordance with the residual meridional circulation, showing negative values during most of the spring and the summer below 92-93 km of altitude, and positive values the rest of the year over the entire altitude range investigated in this work (80–100 km). Agreement with the residual meridional circulation here refers to the fact that if one assumes incompressibility, the divergence of the wind field is zero. Mathematically speaking, this means that the sum of the horizontal divergence and the vertical gradient of the vertical wind is equal to zero. Therefore, a negative horizontal divergence (i.e., convergence in the horizontal plane) must be balanced by a positive vertical gradient of the vertical wind, indicating an upward motion, or upwelling. This upwelling is consistent with the expected behavior of the residual meridional circulation at mesopause altitudes during the summer in high latitudes. Such a clear pattern is not observed in the horizontal divergence over northern Germany (same panel, but in Figure 3), although  $\nabla_H \cdot \mathbf{u}$  still exhibits negative values around the summer mesopause region. The vertical flux of meridional momentum over northern Germany is positive (or poleward directed) during most of the year below ~86 km of altitude and negative (or equatorward) above. The positive values of  $\rho < v'w' >$  extend a little higher during the summer and late autumn, reaching altitudes of about 91 km. A similar behavior of  $\rho < v'w' >$  is observed over northern Norway, although the positive values of the meridional momentum flux are larger and reach higher altitudes (~96 km) than over northern Germany, especially during the summer and the fall transition. Contrary to what happens in the case of  $\rho < v'w' >$ , the relative vorticity behaves very differently over northern Germany and northern Norway, particularly during the summer, when  $(\nabla \times \mathbf{u})_z$  is positive (counterclockwise rotation) at middle latitudes and negative (clockwise rotation) at high latitudes.

In general, positive values of the stretching deformation (third panel, left column in Figures 3 and 4) indicate that big parcels of air stretch from northwest towards southeast, while negative values imply stretching in the southeast/northwest direction. However, a careful look at each individual horizontal gradient of the zonal and meridional winds is needed for a more precise description of the air parcel motion. For example, during summer between 91 and 94 km of altitude over northern Germany,  $u_x$  is positive and  $v_y$  is negative, and thus the stretching  $(u_x - v_y)$  deforms the divergent air parcels in the southwest/northeast direction (see third and fourth panels on the left column of Figure 5). During that same time of the year, but between 91 and 94 km of altitude over northern Norway, the (mostly) convergent air parcels stretch in the northeast/southwest direction given that, although the stretching deformation is negative,  $u_x$  is negative and  $v_y$  is positive (see lower left panels in Figure 6). In the case of the shearing deformation (third panel on the right in Figures 3 and 4), middle and high latitudes present different patterns of this dynamical parameter, especially during the summer, when over northern Germany  $(u_y + v_x)$  changes from

#### MMARIA Norway (1.1.2015-31.12.2024)

Figure 2. Same as Figure 1, but over northern Norway and for the period 1 January 2015 - 31 December 2024. In this case  $(69^{\circ}\text{N})$ , the inertia period is 12.85 hours.

negative to positive as the altitude increases and the opposite is observed over northern Norway. A positive (negative) shearing deformation indicates that the air parcel is under stress that forces it to rotate clockwise (counter-clockwise).




From Figures 5 and 6, it can be seen that the relative importance of divergent and vortical motions at middle and high latitudes over central Europe varies depending on the time of the year and the altitude. In summer, our results at high latitudes (northern Norway) show that the relative vorticity (i.e., vortical motions) dominates below ~91 km of altitude. Above it, vortical motions dominate during the first half of the summerfrom mid-spring on, to then yield the dominant role to the divergent motions. The horizontal divergence also dominates above ~86 km of altitude during the fall and the winter. Below 86 km of altitude, the relative vorticity and the horizontal divergence exchange the dominant role in intervals of time of about one to two months. At middle latitudes (northern Germany), the scenario is different: when one of these two parameters dominates below ~89 km, the other one does so above. This pattern inverts twice during the year, with the divergent motions dominating below ~89 km during winter and half of the equinox transitions, and above ~91 km during the summer. The horizontal divergence also dominates over the relative vorticity during part of the fall transition above 90–91 km of altitude, in coincidence with a reversal of the mean zonal wind from westward to eastward observed above 91 km of altitude. This wind reversal has already been reported in previous studies (e.g., Conte et al., 2018; Jaen et al., 2022). A similar wind reversal has been observed at high latitudes, but for lower altitudes, and has been attributed to the influence of planetary waves and GWs (e.g., Matthias et al., 2015; Espy and Stegman, 2002).

Figure 3. Seven-year climatology of MLT dynamics over northern Germany. From top to bottom: (left) vertical flux of zonal momentum  $(\rho < u'w' >)$ , horizontal divergence  $(\nabla_H \cdot \mathbf{u})$ , stretching deformation  $(u_x - v_y)$  and mean zonal wind (u); (right) vertical flux of meridional momentum  $(\rho < v'w' >)$ , relative vorticity  $((\nabla \times \mathbf{u})_z)$ , shearing deformation  $(u_y + v_x)$  and mean meridional wind (v). The dash-dotted blue contour lines bound the areas of largest year-to-year variability (see text for details). The minor ticks in the x-axis and y-axis appear every 5 days and 1 km, respectively.

## 3.3 Distributions of (4-hour, 4-km) total momentum flux




To infer the role that large-amplitude gravity waves play in driving the momentum fluxes, Figure 7 presents the probability density of the logarithm (base 10) of the total momentum flux over (top) northern Norway and (bottom) northern Germany. The panels on the left correspond to the northern hemisphere's summer season (1 June – 1 September) and those on the right to the wintertime (1 December – 3 March). The TMF is defined as  $\sqrt{(\rho < u'w' >)^2 + (\rho < v'w' >)^2}$ , and the corresponding distributions were constructed considering the 4-hour, 4-km estimates (averaged between 86 and 93 km of altitude, but not in time) of all the summer and winter seasons covered by each MSMR network, i.e., 10 summers and winters for northern Norway, and 7 of each in the case of northern Germany. Following Hertzog et al. (2012), the 95% quantile and the percentage of TMF associated with values larger than this quantile were calculated (in the linear domain) and are indicated in the legend box of each plot. The logarithm (base 10) of these quantiles is marked with the dashed purple vertical lines. As a reference, the normal distributions with mean and standard deviation equal to those of the actual TMF estimates are depicted with the continuous red (northern Norway) and blue (northern Germany) curves. To better assess the intermittency in the TMF, the excess-kurtosis values are indicated with coloured text.

Figure 4. Ten-year climatology of MLT dynamics over northern Norway. From top to bottom: (left) vertical flux of zonal momentum  $(\rho < u'w'>)$ , horizontal divergence  $(\nabla_H \cdot \mathbf{u})$ , stretching deformation  $(u_x - v_y)$  and mean zonal wind (u); (right) vertical flux of meridional momentum  $(\rho < v'w'>)$ , relative vorticity  $((\nabla \times \mathbf{u})_z)$ , shearing deformation  $(u_y + v_x)$  and mean meridional wind (v). The dash-dotted blue contour lines bound the areas of largest year-to-year variability (see text for details). The minor ticks in the x-axis and y-axis appear every 5 days and 1 km, respectively.

The excess-kurtosis of a normal distribution is equal to zero. Thus, a positive value of this statistical moment indicates that the tails of a given distribution are wider, or in other words, that the outliers are more relevant. Figure 7 shows that the excess-kurtosis of the TMF is positive at both locations, but larger over northern Germany than over northern Norway, particularly during the summer: 4.0 against 1.6, respectively. At both locations, the excess-kurtosis is larger in summer than in winter, with respective values of 4.0 and 2.95 over northern Germany and of 1.6 and 1.42 over northern Norway. Concerning the contribution from the strongest 5% of GWs to the total flux, it can be seen that this is also higher at middle latitudes than at high latitudes, with summer again showing a larger difference: 70% contribution to the total flux over northern Germany, while the latter amounts to only 17% over northern Norway. In winter, this difference reduces considerably, to about half: 31% at middle latitudes versus 17% at high latitudes. Overall, the larger values of the excess-kurtosis and the contribution of the strongest 5% of GWs to the total flux observed over northern Germany show that the TMF measured at MLT altitudes over central Europe is more intermittent at middle latitudes than at high latitudes, and that this difference becomes even more significant during the summer.


Figure 5. Wind gradients climatology over northern Germany. From top to bottom: (left column) horizontal divergence  $(\nabla_H \cdot \mathbf{u})$ , stretching deformation  $(u_x - v_y)$ , zonal gradient of zonal wind  $(u_x)$  and meridional gradient of meridional wind  $(v_y)$ ; (middle column) relative vorticity  $((\nabla \times \mathbf{u})_z)$ , shearing deformation  $(u_y + v_x)$ , meridional gradient of zonal wind  $(u_y)$  and zonal gradient of meridional wind  $(v_x)$ ; (right column) difference (in absolute value) between left and middle column parameters. All parameters are in units of  $ms^{-1}km^{-1}$ . The minor ticks in the x-axis and y-axis appear every 5 days and 1 km, respectively.

It is important to mention that the momentum flux estimates here investigated represent a bulk average of waves with periods of 4 hours or less and vertical wavelengths shorter than 4 km. Thus, the summation of contributions from multiple waves with different propagation directions can result in partial cancellation, causing the total measured flux to underestimate the true cumulative momentum. This effect, combined with potential attenuation introduced by the fitting and background wind removal methods, likely reduces the observed amplitude of large TMFs, and might slightly affect the shape of the tail in the distributions presented in Figure 7. These limitations are consistent with those reported in other radar-based studies (e.g., Love and Murphy, 2016), and should be kept in mind when comparing with techniques in which the momentum flux of individual wave packets can be estimated (e.g., Hertzog et al., 2012).

## 4 Discussion



The contribution of large-amplitude gravity waves to the total momentum flux has been investigated before, as it is a helpful quantity to determine how intermittent the gravity wave activity is at different altitudes of the terrestrial atmosphere. A better representation of the intermittency in the gravity wave activity is in turn useful to improve GW parameterization schemes in

## MMARIA Norway / 2015-2024 composite

Figure 6. Same as Figure 5, but over northern Norway. All parameters are in units of  $ms^{-1}km^{-1}$ , and the minor ticks in the x-axis and y-axis appear every 5 days and 1 km, respectively.




global circulation models (e.g., Ern et al., 2022). Using stratospheric balloon measurements and model simulations, Hertzog et al. (2012) showed that the contribution of the strongest 1% of GWs is higher than 20% over mountainous terrain. For MLT altitudes, Conte et al. (2022) used MSMR measurements to estimate the contribution of the strongest 3% of GWs at four different latitudes along the Andes mountain range. They found that the contribution of the large-amplitude GWs is similar between locations that are separated thousands of kilometers, going from about 11% at low latitudes (in Peru) to 13% at middle latitudes (Land of Fire, in southern Argentina). However, they investigated only one southern hemisphere's winter season worth of data. Conte et al. (2023) measured the contribution of the strongest 5% of GWs to the total momentum flux using two winter and two summer seasons worth of data from two MSMR networks located in Peru (low latitudes), and found that the contribution of the large-amplitude GWs to the total flux increases from about 18% in wintertime to 23% during the summer. The increment in the contribution of large-amplitude GWs to the total flux from winter to summer season is also clear at middle latitudes over central Europe. However, the increase measured over northern Germany is much more pronounced than those reported in previous studies, jumping from a 31% contribution in winter to a 70% contribution to the total flux during the summer (see Figure 7). Similarly, the excess-kurtosis measured during summertime shows a considerably larger value compared to that obtained during the winter: 4.0 versus 1.6, respectively. These large values of the excess-kurtosis result from the influence of both large- and small-amplitude GWs.

Figure 7. Probability distributions of the logarithm of the total momentum flux (TMF) over (top) northern Norway and (bottom) northern Germany during the northern hemisphere's (left) summer and (right) winter seasons. The 95th quantile and the percentage of total momentum flux associated with values larger than this quantile are indicated in the legend box of each plot. The logarithm of these values is indicated with the dashed purple vertical lines. The excess-kurtosis is also indicated in coloured text. For reference, the normal distributions (in the logarithmic domain) are depicted with the red and blue curves. The corresponding mean and standard deviation are marked with the dash-dotted and dotted vertical lines, respectively. The plots were made using the 4-hour, 4-km TMF values averaged between 86 and 93 km of altitude, but without averaging in time.

At high latitudes (northern Norway), the contribution from the strongest 5% of GWs is the same during both summer and winter seasons: 17%. However, the excess-kurtosis of the summer distribution is larger than that of the winter (1.6 against 1.42, respectively), which indicates a slightly more significant contribution of the small-amplitude GWs during the summer. The excess-kurtosis is a measure of the relevance that outliers can have. And in the case of the TMF, the outliers can be a consequence of GWs with either large or small amplitudes, while the 95% quantile here analyzed sets a threshold only for the large-amplitude GWs. Given that the latter quantile contributes the same in winter and summer over northern Norway, one can then attribute the increase in the excess-kurtosis to a larger contribution from the small-amplitude GWs. Independently of the amplitude of the waves, both statistical parameters show that the intermittency in the GW activity is higher in summer compared to winter, and that this difference is even more pronounced at middle latitudes over central Europe. Intermittency in the GW activity is the result of variability in both the sources and the medium in which the gravity waves propagate (e.g., Yiğit et al., 2021; Geldenhuys et al., 2021). Non-linear processes can also cause intermittency in the GW activity. In the latter, GWs break and cascade into smaller scales (e.g., Andreassen et al., 1994; Dörnbrack, 1998; Fritts et al., 2009). The breaking of the waves

produces localized forces, which in turn trigger secondary waves with scales that are smaller than those of the primary waves (e.g., Chun and Kim, 2008; Heale et al., 2020). If these processes occur repeatedly enough over several kilometers of altitude, the 28-day momentum flux estimates may capture their mean effect. Following this reasoning, the high intermittency measured during the summer could explain at least part of the year-to-year variability observed at low altitudes in both  $\rho < u'w' >$  and  $\rho < v'w' >$  (see the areas encircled by the dashed-dotted blue contour lines in Figures 3 and 4).






Over northern Norway, negative values of the relative vorticity dominate during the summer over the entire altitude range (see Figure 4 and Figure 6). At high latitudes, the prevailing westward and equatorward winds observed in the mesosphere during the summer months contribute to anticyclonic motions that are characterized by a strong negative relative vorticity (e.g., Harvey et al., 2018). Although the  $(\nabla \times \mathbf{u})_z$  estimate here investigated is associated with mesoscale vortical motions, it exhibits a behavior that is in good agreement with the mentioned planetary-scale anticyclonic motion. In principle, the horizontal wind horizontal gradients used to calculate  $(\nabla \times \mathbf{u})_z$  capture the wind changes that develop in horizontal scales of a few hundred kilometers. And the mean winds that are estimated together with these wind gradients capture the mean planetaryscale dynamics. However, in the same way that some mesoscale variability is present in the mean winds, part of the planetaryscale variability can leak into the wind gradients, and thus the horizontal wind horizontal gradients can still be influenced by the rotation of the Earth, particularly at polar latitudes. Chau et al. (2017) combined twelve years of measurements provided by two monostatic SMRs located in northern Norway in order to investigate, among othersother things, the relative vorticity at high latitudes. Their results show that  $(\nabla \times \mathbf{u})_z$  has the same behavior in both time and altitude as that of the mean zonal wind (see their Figure 7), which is the opposite of what is observed in our results for altitudes higher than 90–91 km. This is due to the different approach they implemented to estimate the relative vorticity and the not so rich limited viewing-angle diversity of their measurements (only two links). Instead of the method described in Section 2 of this paper, they solved for the horizontal wind gradients without taking into account the real shape of the Earth. Once these estimates were determined, they considered the expression  $(\nabla \times \mathbf{u})_z \approx -u_n \cos(\theta_0) + v_x / \cos(\theta_0) + 2u_0 (\tan(\theta_0)/r_0)$  to calculate the relative vorticity. In this expression,  $\theta_0$  is the latitude of the reference point and  $r_0$  is the radius of the Earth at that point, determined using the WGS84 (for more details, see the appendix in Chau et al. (2017)). Although the term that depends on the mean zonal wind is divided by the Earth radius, it has values comparable to those of  $v_x/\cos(\theta_0)$ , and if the wind amplitudes are in the order of 15  $ms^{-1}$  or more, it can easily become the dominant term, hence explaining the similarity in the behavior of their  $(\nabla \times \mathbf{u})_z$  estimate with that of the mean zonal wind. However, as one increases the number of links -(namely, the number of transmitting-receiving pairs), thus improving the viewing-angle diversity, the difference between the two approaches reduces, which underlines the importance of a rich viewing-angle diversity for a reliable estimate of the relative vorticity.

In the case of the results presented in Figure 4, although  $(\nabla \times \mathbf{u})_z$  does not behave in the same way as the mean zonal wind, its partial dependence on planetary-scale vortical motions cannot be discarded. For example, the relative vorticity over northern Norway exhibits the largest year-to-year variability during the winter, which is most likely due to changes in the wind patterns that depend on the development (or not) of strong Polar night Jet Oscillations (e.g., Peters et al., 2018; Conte et al., 2019). Nevertheless, the horizontal divergence estimate presented in that same figure behaves in clear accordance with the

residual meridional circulation, which is driven by mesoscale gravity waves (e.g., Becker, 2012). Hence, it is concluded that the  $(\nabla \times \mathbf{u})_z$  here discussed is the result of both mesoscale and planetary-scale vortical motions.

At middle latitudes, the Coriolis effect is weaker than at polar latitudes. Furthermore, MMARIA/SIMONe Germany has operated with at least six links since 2018 and eleven or more since middle of 2021 (MMARIA/SIMONe Norway operated with four or less from 2015 until middle of 2021). This results in a larger and more homogeneous distribution of meteor detections, which in turn allows for estimates of the relative vorticity that better capture mesoscale vortical motions. The  $(\nabla \times \mathbf{u})_z$  estimate observed over northern Germany shows a different behavior when compared to the climatology over northern Norway, particularly during the summer: positive values (i.e., cyclonic motions) below  $\sim$ 92 km of altitude and negative (i.e., anticyclonic motions) above. Zeng et al. (2024) implemented the VVP method (considering the WGS84) to estimate the relative vorticity from measurements made by a MSMR network recently deployed in central China (around 33°N). They presented results on this and other dynamical parameters between January 2022 and July 2023, finding that the relative vorticity exhibits a similar pattern to the one described here, but that is shifted in time and does not extend as high as over northern Germany:  $(\nabla \times \mathbf{u})_z > 0$  below  $\sim$ 88 km of altitude starting at the end of July in 2022 and at the end of June in 2023. It must be noted that thanks to the larger amount of links, MMARIA/SIMONe Germany has a much richer viewing-angle diversity and thus is better suited to capture mesoscale vortical motions. However, the mentioned discrepancies are most likely due to different dynamics captured by the two MSMR networks, given the distinct geographic areas that their measurements cover.






Still over northern Germany, but in September/October and above ~90 km of altitude, the horizontal divergence dominates over the relative vorticity. This is observed at the time and over the same altitude range where the mean zonal wind changes from eastward to westward, a wind reversal already reported in previous studies (e.g., Conte et al., 2018; Jaen et al., 2022). Espy and Stegman (2002) investigated a similar phenomenon at high latitudes, and attributed it to the effects of both planetary waves and GWs. However, the wind reversal here discussed is not observed at high latitudes. Besides, notice that the westward values of the zonal momentum flux observed above ~85 km during September/October become weaker as the altitude increases (first panel in Figure 3). This feature, in combination with the mentioned dominance of the horizontal divergence over the relative vorticity, suggests that the reversal of the mean zonal wind presented in Figure 3 is mainly the result of momentum redistribution by westward-propagating gravity waves.

Using both MSMR measurements and model simulations, Avsarkisov and Conte (2023) showed that strongly stratified turbulence plays a role as significant as that of mesoscale gravity waves in the summer mesopause region over middle latitudes in southern Patagonia. Figure 5 shows that during the summer at middle latitudes in central Europe, the vortical and divergent motions exchange the dominant role depending on the altitude. Below  $\sim$ 86 km, the relative vorticity dominates over the horizontal divergence, while the opposite occurs above. Around 90 km of altitude, the difference between the absolute values of these two estimates is very small, which indicates that the contribution of vortical and divergent modes is nearly the same around the summer mesopause region over northern Germany. Using one year of measurements from MMARIA/SIMONe Germany, Poblet et al. (2023) found that the divergent and rotational correlation functions of the mesoscale horizontal winds are energetically balanced at those scales over northern Germany. Particularly, averaging  $|\nabla_H \cdot \mathbf{u}| - |(\nabla \times \mathbf{u})_z|$  between 87 and 93 km, the same altitude range that Poblet et al. (2023) investigated, results in values very close to zero, except in May/June and

September/October, when non-negligible negative values are obtained. During those same months, Poblet et al. (2023) reported the rotational component to be larger than the divergent one for the scales that the VVP method can capture, i.e., scales in the order of 300–400 km or more. The approach implemented by Poblet et al. (2023) resolves mesoscales dynamics well and reliably. Thus, the fact that our results are in agreement with theirs allows us to conclude that, provided a rich viewing-angle diversity, the horizontal divergence and relative vorticity estimated with the VVP method are well suited to capture the mean effects of mesoscale motions at MLT altitudes.

## 5 Concluding remarks





This work presents the first climatology of vertical momentum fluxes, horizontal divergence and relative vorticity estimated from multistatic specular meteor radar measurements. The climatology of these dynamical parameters has been investigated in the mesosphere and lower thermosphere over middle and high latitudes in central Europe using 7 years of observations from MMARIA/SIMONe Germany and 10 years of observations from MMARIA/SIMONe Norway, respectively. At high latitudes, the vertical flux of zonal momentum and the horizontal divergence are in agreement with the residual meridional circulation, that is, the former exhibits large positive values (i.e., eastward) below the mesopause, which then become weaker as the altitude increases to eventually become negative in the lower thermosphere. Accordingly, the horizontal divergence shows negative values around the summer mesopause altitudes, which indicates a positive vertical gradient of the vertical wind, or in other words, an upwelling. At middle latitudes, the horizontal divergence is also negative around the summer mesopause region, although the zonal momentum vertical flux does not exhibit such a clear reversal from eastward to westward direction. The relative vorticity shows completely opposite signs during the summer below  $\sim$ 92 km of altitude: positive (counter-clockwise rotation) at middle latitudes and negative (clockwise rotation) at high latitudes. These differences are attributed to the development of upper mesosphere anticyclonic motions during the summer at high latitudes, which are characterized by a strong negative vorticity. Year-to-year variability is observed in all these parameters at both locations, mainly below ~83 km altitude and during the summer over large parts of the altitude range covered by the radar measurements. The relative vorticity, in particular, shows greater year-to-year variability at high latitudes than at middle latitudes, which is likely due to changes in the wind patterns between winters with and without strong Polar night Jet Oscillations.

Comparisons in absolute values of the climatology of the horizontal divergence and the relative vorticity show that the divergent and vortical motions exchange the dominant role depending on the height and the time of the year. However, at summer mesopause altitudes over middle latitudes, the horizontal divergence and the relative vorticity contribute approximately the same, which indicates an energetic balance between mesoscale divergent and vortical motions. During the September/October (fall) transition and above ~90 km of altitude, the horizontal divergence dominates over the relative vorticity. This feature coincides in time and altitude with a reversal of the zonal wind from eastward to westward observed above ~91 km of altitude, suggesting that this change in the direction of the zonal wind, already reported in previous studies, may be the result of momentum redistribution by mesoscale gravity waves. Below ~90 km of altitude, the horizontal divergence also plays the dominant role during winter and both equinox transitions, while above ~91 km it is the relative vorticity the one-that dominates. Vortical

motions also dominate over the divergent ones during the summer, but below ~86 km of altitude. The scenario at high latitudes

is different, with the divergent motions dominating over most of the observed altitude range during winter, and the vortical

motions doing so during the northern hemisphere's late spring and early summer. An interesting pattern is observed during

June/July between ~89 and 93 km altitude: the relative vorticity yields the dominant role to the horizontal divergence, which

makes one speculate on the possibility that secondary gravity waves are being triggered by strongly stratified turbulence in the

high-latitude summer mesopause region. Future studies will be devoted to investigate this puzzling feature.

Finally, the analysis of the total momentum flux distributions reveals that the strongest 5% of gravity waves contribute much

more over northern Germany than northern Norway, particularly during the summer: 70% against 17% contribution to the

total flux. Over northern Norway, the strongest 5% of gravity waves contribute the same during winter and summer seasons

(17%), while over northern Germany the contribution of the strongest 5% of gravity waves during winter is less than half that

of the summer (31% versus 70%, respectively). At both locations, the excess-kurtosis is positive and smaller in winter than in

summer, which indicates a greater influence of the small-amplitude gravity waves during the estival season. The larger values of

the excess-kurtosis obtained over northern Germany (compared to northern Norway) indicate that the small-amplitude gravity

waves are also more significant at middle latitudes, especially during the summer. Simply put, the total momentum flux in

the mesosphere and lower thermosphere over central Europe is considerably more intermittent at middle latitudes than at high

latitudes.

475

485

495

490 Data availability. The meteor radar wind and momentum flux datasets used in this study are available upon request from the corre-

sponding author. The US Standard Atmosphere 1976 density values are available at https://www.ngdc.noaa.gov/stp/space-weather/online-

publications/miscellaneous/us-standard-atmosphere-1976/.

Author contributions. Conceptualization: J. F. Conte. Data curation: T. Renkwitz, R. Latteck, J. F. Conte. Formal analysis: J. F. Conte.

Investigation: J. F. Conte. Methodology: J. F. Conte, J. L. Chau. Software: J. F. Conte. Writing- original draft: J. F. Conte. Writing- review

and editing: J. F. Conte, J. L. Chau, T. Renkwitz, R. Latteck, M. Tsutsumi, C. Jacobi, N. Gulbrandsen, S. Nozawa

Competing interests. C. Jacobi is Editor-in-chief and topical Editor of AnGeo. The authors declare that they have no other competing

interests.

Disclaimer. TEXT

Acknowledgements. The authors thank Nico Pfeffer, Matthias Clahsen, Jens Wedrich and Thomas Barth for their help in maintaining and expanding the MMARIA/SIMONe Germany and MMARIA/SIMONe Norway networks.

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
