# Peer review of "Observing mesoscale dynamics with multistatic specular meteor radars: first climatology of momentum flux, horizontal divergence and relative vorticity over northern central Europe"

_EGUsphere, 2025_

## Author Comment (AC4)

**REVIEWER # 1**

**General comments**

In the manuscript under review, the authors have used multistatic specular meteor radars to show the first climatology of momentum flux, horizontal divergence and relative vorticity over middle and high latitudes in central Europe. For this, 7 years of observations from MMARIA/SIMONe Germany and 10 years of observations from MMARIA/SIMONe Norway, were used. The measurements obtained by multistatic specular meteor radars have proven to be a powerful tool for better understanding of the mesosphere-lower thermosphere dynamics by measuring of parameters that help characterize atmospheric waves and turbulence. The techniques and methods used to achieve the results had already been applied in previous studies and proved to be suitable for the required analyses. The manuscript presentation is clear and the scientific contribution is appropriate for this journal. However, there are a few issues that need to be addressed.

**We kindly thank this reviewer for taking the time to read and revise our manuscript. We hope our responses help to clarify all the concerns and points raised by this reviewer.**

**Specific comments**

Page 2, line 43

In the sentence: "On the other hand, the ability of the GWs to reach the right altitudes necessary", change the word "right' by "suitable" (only a suggestion).

**R. Thanks. We have slightly modified the sentence as follows: "On the other hand, the ability of GWs to reach the altitudes at which they can influence atmospheric processes…"**

Page 8, lines 220-221:

This explanation about highest year-to-year variability in Figures 3 and 4, looks confused, mainly for fluxes and vorticity.

**R. Our apologies if this sentence was not clear. We have rephrased it as follows: "The dash-dotted blue contour lines bound the regions with highest year-to-year variability, i.e., the areas where the 2σ parameter (with σ denoting the standard deviation) reaches the highest values."**

Page 9 lines 245-246

Examining Figure 4 for zonal momentum flux, I see that the summer shows a strong eastward zonal momentum flux above 81 km and below 86 km (not around 81-82 km) of

altitude. Comment the strong westward zonal momentum flux that appear below 81 km during summer.

R. Thanks for this comment. It is not entirely clear to us what may be behind the strong westward zonal momentum flux observed below 81 km during summer at Norway. We think it's the result of anisotropic non-linear gravity waves that propagate with the wind and when breaking accelerate the background zonal wind (notice that those strong westward values approximately coincide with the largest westward values of the mean zonal wind – see bottom left panel in Figure 4). We have observed a similar behaviour at low latitudes (see Conte et al., 2023), but in that case the observations were supported by model simulations exhibiting the same characteristics in the GW drag. Sadly, at the moment we don't have simulations of the same type over Norway. So, our thinking remains only a speculation for now, and we prefer to leave a more thorough investigation of the case for a future study.

Page 10, lines 275-277

In the sentences:

"In summer, our results at high latitudes (northern Norway) show …."

"Above it, vortical motions dominate during the first half of the summer"

Wouldn't it be from mid-spring, instead of summer?

R. Yes, the reviewer is right. Sorry for the mistake. We have modified the sentence as it was suggested.

---

## Author Comment (AC5)

**REVIEWER # 2**

The paper by Conte et al. investigates the effects of mesoscale motions in the MLT and presents the climatology of the transport of vertical momentum fluxes, the horizontal divergence and the relative vorticity. The authors estimate these parameters from multistatic specular meteor radar measurements by applying techniques developed in the past 10 years.

This is a well written paper that deserves to be published. I just have a few observations that should be addressed prior to publication.

**We kindly thank this reviewer for taking the time to read and revise our manuscript. Below, a detailed response to every point raised by this reviewer.**

Regarding the vertical thickness layer of 2 km to estimate horizontal wind. Is there any superposition between adjacent layers?

**R. Yes. Every wind estimate is calculated in an altitude bin of 2 km, centred around the given altitude. For example, for the wind estimate corresponding to 90 km, we use the meteors detected between 89 and 91 km of altitude. Since the bin is shifted 1 km, the next wind estimate will be calculated using the meteors detected between 90 and 92 km. This is a well-established procedure to estimate mean winds using meteor radar measurements (see, for example, Hocking et al. 2001, and Holdsworth et al. 2004).**

page 6, line 166: regarding the gaps in the wind, the authors state that they found 13 hourly gaps in the data and filled them with linear interpolation. How are these gaps distributed? What is the longest sequence of missing data?

**R. The 13 gaps are not consecutive, so the longest sequence of missing data is just one hour. That is what we tried to say with the word "hourly". Nevertheless, we now explicitly say that the gaps are not consecutive.**

Are the units of momentum flux meters*Pascal? The dimension of momentum flux ρ in the metric system is kg*m-3*ms-1*ms-1. How could it end up in m*Pa?

**R. No. The momentum flux units are in milli Pascals (mPa).**

In the discussion session, the authors talk about the "number of links" that the radar systems have. Nevertheless, it is not clear what they are. Please, spend a few words to clarify it.

**R. By "link", we mean every transmitter – receiver pair for a given network. For example, the multistatic meteor radar network we are currently operating in southern Patagonia**

**(SIMONe Argentina) has one transmitter and five receivers. Hence, that network comprises 5 links. We have added to the manuscript a short sentence to clarify this point.**

**Minor**

In text, the authors use the words "fall" and "autumn". I suggest the choice of one of them.

**R. Thanks for this comment, but we prefer to keep using both words to refer to this season of the year.**

page 5, line 357-358: "...thus the horizontal wind horizontal gradients still be influenced by the rotation of the Earth..." I feel some is missing.

suggestion: thus the horizontal wind horizontal gradients still **can/could** be influenced by the rotation of the Earth.

**R. Yes, the word "can" is missing. Thanks for finding this typo. We have corrected it.**

---

## Author Comment (AC6)

**REVIEWER # 3**

Comments on 'Observing mesoscale dynamics with multistatic specular meteor radars: first climatology of momentum flux, horizontal divergence and relative vorticity over central Europe' by Conte et al.

This paper presents long-term observations of mesosphere-lower thermosphere dynamics using specular meteor radar systems whose operation has been extended beyond the standard mono-static design. This development allows new parameters that describe motions within the radar viewing area to be calculated. These parameters are novel and the implications of their magnitudes and variations are still being refined. Here, an extensive climatology for two sites is presented and discussed. The presentation of the data is an important step in our understanding of MLT dynamics. It is difficult to fully judge the veracity of conclusions drawn in relation to the novel parameters but the work provides a substantial building block towards our understanding.

**We kindly thank this reviewer for taking the time to read and thoroughly revise our manuscript. Below, we provide a detailed response to every point raised by this reviewer.**

Specific comments:

The title refers to central Europe. I would recommend it make reference to northern Europe as well.

**R. Thanks. We now say "northern central Europe" in the title.**

Spectral analysis of the winds presented in figures 1 and 2 is used to discuss the nature of tides in the MLT. The calculation of a spectrum of the entire data set has its merits (some of which are unexplored, such as the spectral broadening that can occur due to the presence of various semidiurnal wavenumbers) but it loses sight of seasonal variations in tides. There is potential for more work here by separating the spectra into summer and winter (and perhaps by zooming in on the spectral shape of the peaks associated with the main tidal modes).

**R. Thanks for this suggestion. We prefer to keep this part as it is. Firstly, because we do briefly talk about non-migrating tides, but explain that due to the lack of longitudinal information, we cannot say much about the contribution of different wave numbers. Secondly, the main focus of our manuscript is on mesoscale dynamics at MLT altitudes, so we prefer not to spend too much time on planetary-scale dynamics. We hope the reviewer understands our position. Nevertheless, we have updated the figures showing the spectra to now include the relative amount of (total signal) energy that corresponds to periods shorter than the inertia period, with and without including the dominant tidal waves (i.e., the diurnal and semidiurnal tides).**

A more significant concern with the spectral analysis is the role that diurnal variations in quality of the wind determinations might play. The tidal spectral peaks are a convolution of the true tidal spectrum and the Fourier transform of the daily window function describing the quality of the determination of the corresponding wind component. Certainly, in the case of mono-static meteor wind radars, the quality of the (e.g.) zonal wind determination varies through the day. In this study the amplitudes of the lower period tides could be affected by this convolution process. This brings in to question the detailed discussion of the 6, 4.8 and 4 hour tides. The paper would not be adversely affected by leaving out these discussions but some consideration of the daily variation of the quality of the determinations of U and V would be an important addition to the paper and our knowledge of the multistatic radar method (or perhaps this has been done elsewhere and a reference can be provided).

R. Thank you for this comment. We are aware that shorter-period tides like 6, 4.8, or 4 hours are vulnerable to being distorted by a diurnal variation in the data quality. However, the spectral contamination that may come from diurnal variation in the data quality should affect similarly both the U and V components, assuming the data quality varies similarly for both. And indeed, the data quality is the same for both components because

a. The errors in the mean winds and the tidal estimates are pretty much the same in both components (and small, not more than 5%). We do not see that one component (either in the mean winds or in the tides) is noisier than the other.

b. The multistatic configuration of both networks allows for a more homogeneous distribution of meteor counts throughout the day. The hourly detection rates are sufficiently uniform to reliably extract the mean winds and tides without favouring any particular direction.

c. In the case of the 4-hour tide over Norway, we only see a peak in U (and not in V) consistently through many days and years. The peak remains always above the noise level and is phase coherent. For these reasons, we think this is a real tidal wave, but with a strong anisotropy.

We have rephrased the whole paragraph accordingly. The new paragraph now reads as follows

"... These features are consistently observed across many days and years. In the case of the 4-hour tide, the spectral peak remains above the noise level and exhibits phase coherence (not shown here), suggesting that it represents an actual tidal wave rather than an artifact arising from daily variability in data quality. Furthermore, the errors in the mean winds and tidal estimates are comparable between both components, with no indication that one component (either in the mean winds or in the tides) is noisier than the other. In addition, the multistatic configuration allows for a more homogeneous distribution of meteor counts throughout the day, and the hourly detection rates are sufficiently uniform to reliably extract the tides without favoring any specific direction. Based on these considerations, it is concluded that the short-period tidal peaks are indeed actual tides and not artifacts introduced by data quality issues. However, understanding, for instance, why a short-period tidal wave propagating in the zonal direction may exhibit a much smaller meridional component is left for future investigations."

The discussion around line 205 relating to the role of the Coriolis effect is vague and not convincing. In the context of the tides being discussed, it would be more relevant to consider the latitudinal variation of the Hough 'functions' for U and V.

**R. Thank you for this comment. Sorry if this was not clear, but we do not pretend to discuss the tides in this part of the manuscript, but rather address the similarity between the slopes at mesoscales. We have modified the text around line 205 to better explain our point, which now is also accompanied by the new Figures 1 and 2, which show the relative energy percentage before and after removing the tidal peaks.**

Section 3.3:

Does the analysis allow large MF (or TMF) values or are they attenuated by it? Given that there is consideration of GW intermittency in the paper, some discussion on the analysis method's ability to reliably capture the amplitude of a large GW unattenuated is important (e.g. around line 291).

There is a paper by Love and Murphy (2016) doi:10.1002/2016JD025627 that is relevant here and should be included in the discussion.

A concern about these considerations of the momentum flux distribution is that the method described here does not extract the MF of individual wave events to the extent that the super pressure balloon work of the various Hertzog et al. papers and the radar work of Love and Murphy (2016). Here, the total MF, which is potentially a combination of multiple waves, is measured. Noting that MF is a vector quantity, the total MF will likely be smaller than the MF of the component waves. This factor, and the potential attenuation of the MF due to the fitting method mentioned above, need to be discussed when presenting the distributions in figure 7 because they will affect the shape of the tail. The results are of interest but differences to other observations (e.g in the amount of MF held by the large waves) could be explained in this way.

**R. Thank you for this helpful comment. We agree that the method to estimate momentum fluxes from Doppler residuals averaged over time and height does not isolate individual gravity wave packets as done in the studies by Hertzog et al. 2012. However, we would like to clarify that the radar work by Love and Murphy (2016) also employs bulk, time-averaged MF estimates rather than event-resolved flux extraction. Thus, the comparison to their method as an example of individual wave MF extraction is not entirely accurate.**

**However, we are fully aware that our momentum flux estimates may suffer from cancellation effects when multiple waves with different propagation directions are averaged together. This vector summation combined with the attenuation introduced by our fitting method may result in the reduction of the apparent amplitude of large MF events and influence the shape of the distribution tail, as shown in our Figure 7. We have now made this limitation explicit in the revised manuscript.**

The contours showing 2 sigma variation in the climatology that are included in many of the plots are a good addition to them.

**R. Thanks.**

Technical comments:

L13 – suggest replace 'balanced' with 'equal'

**R. Thanks. However, we think the word "balanced" is more appropriate here.**

L28 – insert 'with' after 'hours,'

**R. Done. Thank you.**

L29 – insert 'typically' after 'wavelengths'

**R. Done. Thanks.**

L43 – suggest delete 'only'

**R. Thanks. The word "only" is important in this sentence so we prefer to keep it.**

L42 – suggest replace 'the horizontal' with 'GW horizontal'

**R. Thanks. It's done.**

L117 – Is the mathematical nomenclature correct here? It looks like a matrix equation. I think there should be a dot between the RHS terms and perhaps add commas after each term in the vertical stack between the large parentheses.

**R. Yes, you are right. We have now written it correctly, as you suggested.**

L130 – the explanation of the method described in equation (3) and this line is confusing. Given that the vector expression (u' dot k) yields a number, how is its square not also a number. And yet, Reynolds stress terms result. Can you please clarify and/or rewrite this?

**R. Yes, you are correct. The result will be a number, but that number is a linear combination of the six components of the Reynolds stress tensor, i.e.,**

$$(u'^2)k_x^2 + 2(u'v')k_xk_y + 2(u'w')k_xk_z + 2(v'w')k_yk_z + (v'^2)k_y^2 + (w'^2)k_z^2$$

**So, theoretically, with 6 Doppler shift measurements we can construct a linear system of equations, and solve for the six components of the stress tensor. We actually use at least**

**40 meteor detections for more reliable estimates. Hopefully, now it is clear how we did the estimation.**

L161 – Suggest delete 'Besides'

**R. Thanks. It's done.**

L210 – Suggest replace 'with horizontal scales where the Coriolis effect has little to no impact (i.e. mesoscales)' with 'that are not harmonics of the solar forcing'

**R. Thanks. But we are referring not only to scales corresponding to waves with periods that are not harmonics of the solar forcing, so we prefer to leave the sentence as it is.**

L227 - Suggest delete 'Besides'

**R. Thanks, but we prefer to keep the word "Besides" in this sentence.**

L236 – When introducing figures 5 and 6, the authors should state why the various parameters on this line are of interest.

**R. Thanks. We state this at the beginning of that paragraph, when we say "To assess the relative importance between mesoscale waves and strongly stratified turbulence, and also have a better picture of how big volumes of air move at MLT altitudes, Figure 5 and Figure 6 respectively present…"**

L250 – Please provide more explanation of what is meant by 'in accordance with the residual meridional circulation'. Do you mean it coincides with its variations or are you suggest a relationship?

**R. Sorry if this was not clear. We have now added the following paragraph, which we expect makes things clearer:**

**"Agreement with the residual meridional circulation here refers to the fact that if one assumes incompressibility, the divergence of the wind field is zero. Mathematically speaking, this means that the sum of the horizontal divergence and the vertical gradient of the vertical wind is equal to zero. Therefore, a negative horizontal divergence (i.e., convergence in the horizontal plane) must be balanced by a positive vertical gradient of the vertical wind, indicating an upward motion, or upwelling. This upwelling is consistent with the expected behavior of the residual meridional circulation at mesopause altitudes during the summer in high latitudes."**

L336 – insert 'a' before 'consequence'

**R. Done. Thanks.**

L357 – insert 'can' after 'gradients'

**R. Done. Thanks.**

L359 – change 'others' to 'other things'

**R. Thanks. It's done.**

L362 – change 'not so rich' to 'limited'

**R. Thanks. Done.**

L363 – Please explain what you mean by a 'link' when you introduce this term here.

**R. Thanks. We now say "the number of links (namely, the number of transmitting-receiving pairs)".**

**We expect this clarifies what we mean by "link".**

L399 – insert 'km' after '85'

**R. Done. Thanks.**

L445 – delete 'the one'

**R. Thanks. Done.**